# Health Risk Assessment in Children Occupationally and Para-Occupationally Exposed to Benzene Using a Reverse-Translation PBPK Model

**DOI:** 10.3390/ijerph20032275

**Published:** 2023-01-27

**Authors:** Kristal Pech, Norma Pérez-Herrera, Ángel Antonio Vértiz-Hernández, Martín Lajous, Paulina Farías

**Affiliations:** 1Instituto Nacional de Salud Pública, Cuernavaca 62100, Mexico; 2Laboratorio de Enfermedades Crónicas y Degenerativas, Unidad Interinstitucional de Investigación Clínica y Epidemiológica, Universidad Autónoma de Yucatán, Mérida 97000, Mexico; 3Coordinación Académica Regional Altiplano, Universidad Autónoma de San Luis Potosí, San Luis Potosí 78300, Mexico; 4Department of Global Health and Population, Harvard T. H. Chan School of Public Health, Boston, MA 02115, USA

**Keywords:** risk assessment, PBPK, children, benzene, leukemia, lymphopenia

## Abstract

Benzene is a known human carcinogen and one of the ten chemicals of major public health concern identified by the World Health Organization. Our objective was to evaluate benzene’s carcinogenic and non-carcinogenic health risks (current and projected) in highly exposed children in Yucatan, Mexico. Benzene exposure was estimated through a reverse-translation, four-compartment, physiologically based pharmacokinetic model (PBPK) based on previously performed urine trans, trans-muconic acid (benzene metabolite) determinations. Using a risk assessment methodology, the carcinogenic and non-carcinogenic risks of benzene were estimated for 6–12-year-old children from a family of shoemakers. The children’s hazard quotients for decreased lymphocyte count were 27 and 53 for 4 and 8 h/day exposure, respectively, and 37 for the projected 8 h/day exposure in adults. The risks of developing leukemia were 2–6 cases in 1000 children exposed 4 h/day; 4–10 cases in 1000 children exposed 8 h/day, and 2–9 cases in 1000 adults with an 8 h/day lifetime exposure. Children in Yucatan working in shoe-manufacturing workshops, or living next to them, are exposed to benzene concentrations above the reference concentration and have unacceptably high risks of presenting with non-carcinogenic and carcinogenic hematologic symptoms, now and in the future. Interventions to prevent further exposure and mitigate health risks are necessary.

## 1. Introduction

Benzene is considered one of the top ten chemicals of concern by the World Health Organization (WHO). It is a high-production-volume chemical (>1000 metric tons/produced per year) and a group 1 human carcinogen according to the International Agency for Research on Cancer [1].

Several studies have shown that benzene exposure can cause hematologic, hepatic, respiratory, and pulmonary adverse health effects in children [2,3]. Unlike adults, children have physical, physiological, and behavioral characteristics that may make them more vulnerable to benzene exposure and prone to an increasing number of health effects, some irreversible [4]. A meta-analysis suggested that hemotoxicity through bone marrow injury, significantly reducing red and white blood cell counts, may be benzene’s most significant health effect in children [2]. Although white blood cell count can fluctuate substantially without adverse health effects, absolute lymphocytic count is the most sensitive measure of benzene’s hemotoxicity, and this can play a role as a sentinel effect for several of benzene’s associated hematological adverse effects [1]. If the decreased lymphocyte count reaches the age-dependent thresholds needed to be classified as lymphopenia, which can occur with prolonged and high exposures to benzene, it may compromise the immune response; this, in turn, can increase the risk of acute myeloid leukemia (AML) [5]. Several studies have documented this increased risk of leukemia in children exposed to benzene [3,6,7]. the summary relative risk for leukemia in a 20-study meta-analysis of children exposed to benzene was 1.96 (95% confidence interval: 1.53, 2.52) [8].

Aside from an altered immunity, in vivo and in vitro toxicological studies evaluating pharmacokinetics, hematologic toxicity, cytogenicity, genotoxicity and carcinogenicity suggest other possible mechanisms of benzene’s carcinogenicity [9,10], such as oxidative stress. It is also believed that benzene is not a direct-acting mutagen, but that its effects arise from its metabolites, such as 1,4-benzoquinone and muconaldehyde [11]. Increased leukemogenesis risk in children could be expected due to their hematopoietic cell populations undergoing differentiation and maturation [1].

Deleterious hematopoietic outcomes can have important health implications. For instance, Mexico’s National Health and Nutrition Survey 2018–19 estimated an anemia prevalence in school-aged children of 19.2% (95% CI 17.3–21.2) in the whole country, and of 22.9% (95% CI 20.1–25.9) in the southern region of the country (where the study site is located in the state of Yucatan) [12]. Regarding carcinogenic effects, acute leukemia is the most common childhood cancer. In Mexico, during 2020, 1258 incident cases were reported in men and 1031 in women under 20 years of age [13]. According to data from the Institute for Health Metrics and Evaluation (IHME), during 2019, mortality attributed to leukemia in children aged five to fourteen years for both sexes in Yucatan, representing 12.8% of all deaths in this age group [14]. In addition, the Merida (Yucatan’s state capital) Population-Based Cancer registry reported 103 incident cases of leukemias, myeloproliferative disease and myelodysplasias during the period 2015–2018 in children aged 0–19 years [15]. The incidence and mortality derived from these diseases is likely influenced by environmental hazards, which the evidence of a steady increase in childhood leukemia points to, especially in Hispanics [16]. Studies of exposure to environmental hazards and leukemia are lacking in Yucatan’s child population. However, in the city of Ticul, Yucatan, trans-trans muconic acid (tt-MA), a metabolite of benzene, has been previously detected in urine samples of children living in or next to households with shoe workshops [17]. Handcrafted shoe production, a characteristic activity in Ticul, is conducted under unsafe working conditions and without the necessary protective measures. In addition, it is socially acceptable for children to participate in shoemaking [17,18]. In the production process, glues, solvents, and paints (all containing benzene) are used; hence, children are potentially exposed to benzene, occupationally and para-occupationally [19]. Understanding the timing and sources of benzene exposure in Ticul is challenging because children can also be exposed during gestation via maternal occupational or para-occupational exposure [20], as well as via other sources of exposure, including smoking or biomass burning [1].

Based on high benzene exposure estimates in children in Ticul, a risk assessment was deemed necessary to assess the nature and magnitude of potential health risks in this population. Thus, we evaluated the current and future risk of developing lymphopenia and leukemia due to benzene exposure in children aged 6 to 12 years living Ticul, Yucatan under different exposure scenarios. 

## 2. Materials and Methods

### 2.1. Study Site

The city of Ticul is located 85 km south of Merida, Yucatan, which in turn is located in the northern part of the Yucatan Peninsula in southeastern Mexico. It occupies an area of 339.9 km^2^ and has a total population of 40,495 inhabitants, of which approximately 70% is indigenous [21].

There are about 234 shoe workshops in Ticul, with an estimated production of 345,000 pairs of shoes per year, according to the national statistical directory of economic activities (DENUE, 2020) distributed throughout the municipality (Figure 1) [22]. Some shoe workshops are located inside homes and children who live there often participate in some shoemaking processes [17]. In Ticul, there are other sources of exposure to benzene, such as vehicular traffic, biomass burning, and gasoline stations [17].

### 2.2. Study Population

The study population consists entirely of children in Ticul that are occupationally and/or para-occupationally exposed to benzene. According to the 2020 population census, in Ticul, there are 7041 children aged 5 to 14 years old, and the birth rate of the local reproductive-aged women (15 to 45 years old) is two children on average [21]. Assuming that in each shoe workshop (n = 234) there are two children, 7% of Ticul’s 5- to 14-year-old children (468 children) may be potentially exposed to benzene. 

### 2.3. Exposure Assessment

In order to compare the study population’s benzene exposure with benzene’s reference concentration for a risk assessment, it was necessary to measure the concentration of benzene in the air. Since air benzene measurements were not available for our population, we estimated them from the urine tt-MA concentrations found in a previous study of children in Ticul (2019) [17]. Public invitations were made to parents of school-age children attending one of the three schools closest to the sites with the greatest environmental risks to public health, including footwear manufacturing, as well as agricultural and municipal garbage areas. The aforementioned investigation included a sample of 6- to 12-year-old children of shoemakers, potters or farmers family. One child per household was included for a total of 41 children who met the inclusion criteria. A urine sample from the participating children was collected to determine tt-MA levels based on the method described by Ducos et al. (1992) [17,23].

For the current study, we excluded 23 children with tt-MA concentrations below the limit of quantification (0.06 µg tt-MA/L) and 3 children with abnormal albumin/creatinine ratio (3 children) because a normal renal function is assumed in the toxicokinetic model [24]. The final population used for determining benzene exposure was 15 children.

#### 2.3.1. Reverse-Translation Toxicokinetic Model

Benzene air concentrations were estimated using a reverse-translation physiologically based pharmacokinetic model (PBPK) with the input of biological data corresponding to children (Figure 2), following the general equation according to the model of Majumdar et al. (Equation (1)) [24]. Four compartments were considered: richly perfused tissues, fat, poorly perfused tissues, and liver. The variables considered were tissue volume, partition coefficient, flow velocity, maximum metabolic capacity, Michaelis Menten constant, and half-life time for benzene; their values were obtained from studies by Valcke and Krishnan [25,26]. Urinary volume and frequency, as well as the urinary creatinine of 6 to 12 year-old children, were also considered [27], as well as data on age, weight, height, and tt-MA levels reported in the Ticul study by Pérez-Herrera et al. (2019).

In Equation (1), Cl is the concentration of benzene (mg/min): (1)$$\;\;Cl=\frac{Q_{i}P_{i}C_{art}t}{\left(V_{i}P_{i}\right)+\left(Q_{i}t\right)}$$
where *Q_i_* is the flow rate (L min^−1^), *P_i_* is the partition coefficient, *C_art_* is the arterial concentration of benzene (µg L^−1^), *t* is time (min), and *V_i_* is the volume of each compartment (L).

#### 2.3.2. Exposure Scenarios Evaluation

A benzene inhalation exposure scenario evaluation was performed using the procedure described by the U.S. Environmental Protection Agency (EPA) [28]. Considering information previously collected on the labor situation in Ticul, the assumed conditions necessary to cause benzene exposure were a Monday to Saturday working week in a shoe workshop and a two-week vacation per year, totaling 300 days per year of exposure. Additionally, we assumed that children stated working in the shoe workshop at around six years of age since this was the age when exposure was considered to begin. Three exposure scenarios were created by the authors: scenario a—shoemaker family children who are para-occupationally exposed to benzene 4 h per day for 300 days per year for 6 years because their living spaces are shared with the workshop, but they spend time elsewhere, such as at school or spending time playing outdoors; scenario b—children who are occupationally exposed 8 h per day for 300 days per year for 6 years by working 4 h/day in the workshop but spend time (4 h/day) in indoor spaces shared with the workshop; scenario c—projected benzene exposure in adulthood for children exposed from birth to 70 years of age, for 8 h a day for 300 days a year.

Equation (2) shows our exposure assessment model, where CE (mg/m^3^) is the chronic exposure averaged over the exposure time for non-carcinogenic hazards or over a lifetime for carcinogenic hazards.
*CE = (C × ET × EF × ED)/1440 × AT*(2)
where *C* is the air concentration of benzene expressed in mg/m^3^; this was the concentration estimated with the reverse-translation PBPK model (5.85 mg/m^3^). *ET* is the exposure time (min/day); we used a period of 4 and 8 h which gives an exposure time of 240 and 480 min/day, respectively. *EF* is the exposure frequency (days/year); the exposure frequency of 300 days/year was for the three scenarios taking into account non-working days. *ED* is the exposure duration (years), and *AT* is the average exposure time (calculated by multiplying *ED* × 365 days and expressed in days). The parameters and values used for these scenarios are summarized in Appendix A. The exposure scenarios calculations were performed using The US Environmental Protection Agency’s (EPA) ExpoFIRST tool [28].

### 2.4. Risk Assessment

The non-carcinogenic risk (decreased lymphocyte count) was assessed with the hazard quotient (*HQ*), which expresses the ratio of the chronic exposure (*CE*) and the reference concentration (*RfC*), defined as the estimate of a continuous inhalation exposure for the human population that is unlikely to present an appreciable risk of harmful effects over a lifetime [29], as described below.
*HQ = CE/RfC*(3)

If the resulting *HQ* is less than one, no adverse health effects are expected as a result of exposure. For benzene, the *RfC* is 0.03 mg/m^3^, and its corresponding critical effect is a decreased lymphocyte count according to IRIS, EPA [29].

The carcinogenic risk for leukemia was estimated by multiplying the *CE* by the inhalation unit risk (*IUR*) for benzene air concentration; the unit risk is the upper-bound excess lifetime cancer risk estimated caused by continuous exposure to an agent at a concentration of 1 (µg/m^3^)^−1^ in breathed air, as described below.
*Increased Cancer Risk (ICR) = CE × IUR*(4)
where *IUR* for leukemia development can range from 2.2 × 10^−6^ to 7.8 × 10^−6^ (µg/m^3^)^−1^ and both values were used to estimate the risk scenarios [29].

## 3. Results

The mean (0.83 ± SD) urinary tt-MA concentration of 15 children exposed to benzene and aged 6 to 12 years, was 0.81 mg/L, and the estimated mean (13.9 ± SD) concentration of benzene in the air of the shoe workshops adjacent to their living spaces and/or where they work, was 5.85 mg/m^3^. The estimated average daily exposures considering the previously mentioned air benzene concentration for the three scenarios addressed in this study, as well as the respective non-carcinogenic and carcinogenic risks estimated based on those exposures, are shown in Table 1. The average chronic daily exposure to benzene was above its reference concentration (0.03 mg/m^3^) in the three scenarios; thus, the HQ for a decreased lymphocyte count was also >1 in all exposure scenarios.

The risk of developing leukemia ranged from 2 to 6 cases in 1000 children exposed to the aforementioned air benzene concentration for 4 h/day; 4–10 cases in 1000 children exposed during 8 h/day, and 2–9 cases in 1000 adults with an 8 h/day lifetime exposure. Considering there are approximately 468 children under either of these exposure conditions, 1–3 cases of leukemia were expected if they were all exposed for a maximum of 4 h/day, and 2–5 cases were expected if they were all exposed for a maximum of 8 h/day.

## 4. Discussion

Our results suggest that children exposed to benzene from shoemaking activity in Ticul are at risk of presenting non-carcinogenic and carcinogenic hematopoietic effects, now and in the future, if exposure persists. Even under the best case scenario, exposure to benzene was almost 27 times above the exposure dose at which no health effects would be expected to develop from daily exposure over a lifetime.

Epidemiological studies have consistently found an association between benzene exposures (presumably to lower concentrations than those found in the present study) and decreased white blood cell and absolute lymphocytes counts in children of similar ages [30,31,32]. This exposure to benzene is associated with altered immune status due to a decrease in white blood cell counts [33]; it is important considering the impact it may have on the complication of infectious diseases under Ticul’s situation of vulnerability, as well as conditions such as the COVID-19 pandemic [34,35].

It is important to emphasize that decreased white blood cell and lymphocyte counts are the first benzene-associated outcomes to appear; however, there is evidence that exposure to benzene is also associated with other non-cancerous effects, including hematological effects, such as anemia and thrombocytopenia [2]. This is a situation worth considering since the prevalence of anemia is higher in the southern part of the Mexico, and even more so in rural areas such as Ticul [36]; therefore, benzene exposure could be a factor influencing the incidence and mortality from this condition or conditioning its long-term consequences such as impaired growth, cardiac function, cognitive development or other poorly described consequences [37]. Occupational exposure to benzene above 2 ppm is reported to cause significant decreases in white blood cell count [38].

Regarding benzene’s carcinogenic risk, the three exposure scenarios were more than one order of magnitude above the upper limit of what is considered an acceptable risk for the general population (1 in 10,000) [29]. A risk of leukemia ranging from 2 to 6 cases in 1000 children was found for those exposed 4 h/day. Notably, scenario a is conservative in the sense that it considers the time children spend outside the home playing or going to school; however, it does not consider situations such as those experienced during the COVID-19 epidemic, in which children spent more time inside their homes. Scenario b assumes an occupational exposure of 8 h (parents’ working day); however, it should be noted that not all children work in shoemaking and that perhaps most of them are exposed to benzene because they live in homes where this activity is performed. Therefore, the exposure is mainly para-occupational but still very high due to their living conditions [17]. Finally, the risk of scenario c representing the future of children that are exposed from birth, due to living in homes where adults practice shoemaking, is still unacceptable, ranging from 2 to 9 subjects exposed between birth and adulthood.

This information is extremely important since it is known that acute lymphoblastic leukemias (ALL) represent 50% of childhood cancer cases and are the primary cause of death in children aged 5 to 14 years in Mexico [39]. Although the causes of leukemias are not fully understood, there is a genetic component, and an increasing number of studies are attributing the incidence of leukemias to environmental and preventable factors, such as exposure to benzene [16].

PBPK modelling is a relatively novel tool used in risk assessments, whose main advantage is its predictive capacity since it estimates the internal dose. This model reduces the uncertainties that would exist in the case of using assumptions taken from other scenarios, such as those of Ticul, so the main advantage of our study is that we use data from the exposed population [40].

The estimated benzene concentration from the PBPK model (5.85 mg/m^3^ =1.83 ppm) that we reported is almost two-fold OSHA’s maximum permissible level for occupational exposure to benzene, which should not exceed 1 ppm for an 8 h workday. However, this is similar to studies of other occupational settings. For example, in Mexico, the air benzene concentration reported in a shoe factory in Guanajuato was 3.7 mg/m^3^ [41]; in China, airborne benzene levels of 4.8 mg/m^3^ were reported in a shoe factory in 2006 [42]; and a study conducted in Iran reported benzene concentrations of 3.51 mg/m^3^, 4.37 mg/m^3^, and 4.85 mg/m^3^ in a shoe factory during the months of October, November and December, respectively [19].

It is important to emphasize that children are more sensitive to toxic chemicals than adults due to their body weight, immature metabolic pathways, the disruption of early developmental processes, and a longer period in which chronic diseases can develop [4].

The limitations of this study include the uncertainty inherent in each step of the risk assessment, particularly regarding the assumptions of the PBPK model. The uncertainty factors (values) considered for this risk assessment were: effect level extrapolation because the decreased white cell count is already the presence of an adverse effect (3), variability in the response to exposure among human beings in order to ensure the protection of sensitive individuals (10), an extrapolation for subchronic–chronic exposure (3), and database deficiencies since neither two-generation reproductive nor developmental toxicity has been studied and accounted for (3) [29]. These assumptions were as follows: (a) the human subject is exposed only by inhalation to a constant level of benzene in air for eight hours (working day); (b) the benzene level in the subject’s blood instantaneously reaches a steady-state concentration; (c) the arterial concentration of benzene in different body compartments occurs homogeneously; (d) a certain amount of benzene is continuously metabolized in the liver, and metabolites dissolved in urine accumulate in the bladder; (e) the average time between successive micturitions (emptying of the bladder) is three hours [24]. The sample size and composition used for the PBPK model exposure estimation could be an additional limitation of our risk assessment. The children included in the model might not be representative of all the benzene-exposed children, but excluding more than half of the original 41 children increased the probability of representativeness, and we have no reason to believe that the remaining 15 are any different from all children from shoemaker families in terms of their biokinetic parameters. Furthermore, the characterization and validation of the PBPK model evaluates data such as formulas and pharmacokinetic and physiological parameters. The original study design was a quota model, and according to the guidelines of the Secretary of Health, for research in humans and in drug bioavailability studies, the number of individuals must be ≥12 (NOM-117-SSA1-2013) [43,44]. Since exposure was determined from a single urine sample of each of the included children, we are assuming it was representative of a consistent pattern of exposure. Due to the previously described conditions of Ticul, our study focused on evaluating the occupational exposure derived from shoemaking activities. Therefore, we excluded tt-MA concentrations that were found below the limit of quantification—assuming that tt-MA concentrations higher than this limit came from this occupational exposure and not from other sources (traffic, biomass, gasoline stations)—with low mean benzene concentrations compared to those that we obtained [45,46,47].

The exposure and the risk assessments in this study might be underestimated since prenatal exposure was not accounted for; this affects both the sum and the timing of the exposure during a critical window of development. Maternal occupational exposure to benzene during pregnancy has been significantly and repeatedly associated with an increased risk of leukemia in children [48,49]. Prenatal exposure was likely present in the studied children, but it was not considered in this study due to a lack of data.

Finally, it is important to consider that the sources of exposure to benzene (solvents, glues, and paints) are mixtures of substances, so children are exposed to other toxicants that are not accounted for, such as toluene, and this could increase health risks. In addition, there are other identified contaminants in Ticul, such as pesticides, arsenic, chromium, and mercury [18]. An additive or a synergistic effect should also be considered in future studies.

## 5. Conclusions

The living and working conditions of children from shoemaker families in Ticul, Yucatan led to a benzene exposure associated with a high risk of developing both non-carcinogenic and carcinogenic hematological effects, now and in the future. The assumptions made to estimate the exposure and, consequently, the risks, our conservative approach, and the lack of consideration of potentially synergistic exposures to other hematotoxic agents may have led to an underestimation of such risks. Furthermore, other health risks associated with benzene exposure (nephrotoxicity, neurotoxicity, and pneumotoxicity) and the conditions of vulnerability in which the study population lives were not accounted for. Care should be taken not to disregard a much larger health impact for the studied children than that reflected in our results.

Benzene-associated lymphopenia and leukemia are preventable; given children’s occupational and para-occupational benzene exposure situation in Ticul, primary, secondary and tertiary preventions are urgently required. Primary prevention, consisting of preventing the exposure of newly pregnant women and children who reach a locally acceptable working age (six years), is of paramount importance to avoid both critical windows and concentrations of exposure. It also calls for the prevention of any further exposure of already-exposed children, in order to decrease their risk as much as possible. Secondary prevention in children who are currently or have already been exposed to high concentrations of benzene, and whose risks are estimated in this study, require a biological screening, both for exposure biomarkers, such as urine tt-MA levels, as well as blood biometry for an early detection of abnormalities in any cell line (lymphoid, myeloid). A tertiary prevention would involve following up previously exposed children for the rest of their lives and being prepared for the early detection and treatment of hematologic diseases, especially leukemia. In addition, the use of personal protective equipment should be promoted for exposed workers; workshops that are located inside homes should be relocated, isolated, and ventilated; and children should avoid participating in any shoe-manufacturing processes.

This study’s results can support a tailored risk assessment for Ticul to increase the population’s and health authorities’ compliance with the three levels of prevention and their success. Our results can provide an idea of what children in similar circumstances in other developing countries are facing. More studies in which air benzene concentrations are measured in shoe workshops would be useful to corroborate the exposure estimated in this study. Additionally, the follow-up of this population of exposed children would be useful study subjects, not only for their personal benefit regarding different levels of prevention, but also to aid a much-needed and more complete understanding of the dose–response curve of benzene in children.

## Figures and Tables

**Figure 1 ijerph-20-02275-f001:**
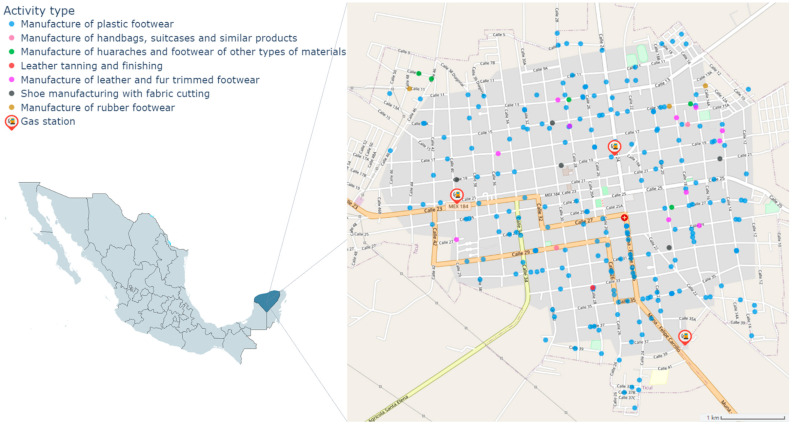
Map of Ticul with the locations of shoe workshops.

**Figure 2 ijerph-20-02275-f002:**
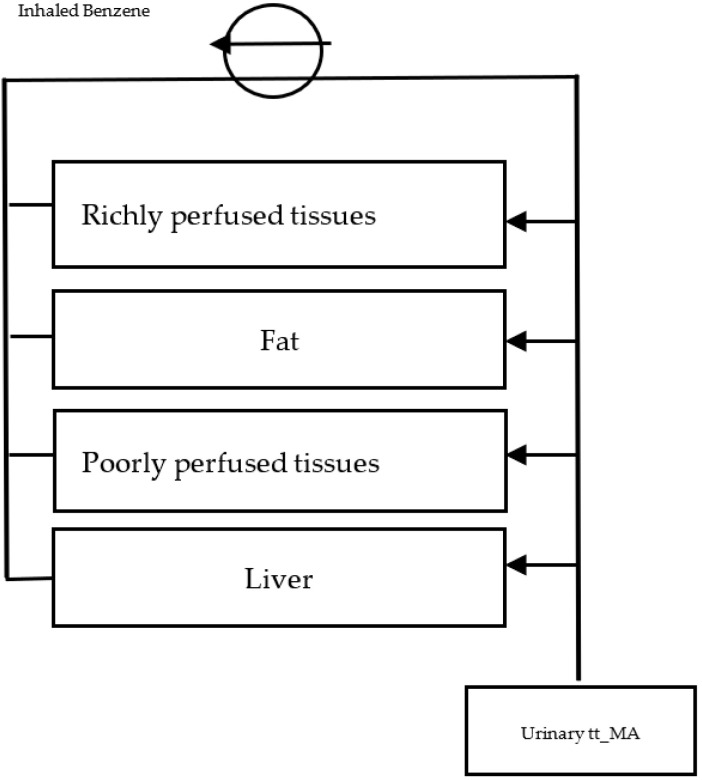
Reverse-translation PBPK model.

**Table 1 ijerph-20-02275-t001:** Carcinogenic and non-carcinogenic risk from exposure to inhaled benzene in children from Ticul.

Exposure Scenario	Chronic Exposure (CE) (mg/m^3^)	Hazard Quotient (EC/RfC ^a^)	Carcinogenic Risk (CE.IUR ^b^)
(a) Children: 6 to 12 years old exposed 4 h/day	0.80	26.7	2–6 × 10^−3^
(b) Children: 6 to 12 years old exposed 8 h/day	1.60	53.3	4–10 × 10^−3^
(c) Adults: projection for 8 h/day exposure during 70 years of life	1.11	37	2–9 × 10^−3^

^a^ RfC = reference concentration (0.03 mg/m^3^). ^b^ Inhalation unit risk (IUR) = 2.2 × 10^−3^–7.8 × 10^−3^ (mg/m^3^)^−1^.

## Data Availability

The data presented in this study are available on request from the corresponding author. The data are not publicly available due to privacy issues.

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
