# Peer review of "Health Risk Assessment in Children Occupationally and Para-Occupationally Exposed to Benzene Using a Reverse-Translation PBPK Model"

_ijerph, 2023, doi:10.3390/ijerph20032275_

Round 1

Reviewer 1 Report

The study is very interesting and the methodology is well described. Yet, small sample size is the major consideration. Therefore, if the sample is not representative for the child population at risk, the results are meaningless. The authors need to explain how the sufficient sample size was determined. Another thing that has to be described separately is the sampling strategy. I understand that this study relies on the earlier collected data, but these issues need to be addressed. Otherwise, this manuscript is nothing but a modeling exercise.

Author Response

Reviewer 1

The study is very interesting and the methodology is well described.

Thank you, we address each comment below.

  1. Yet, small sample size is the major consideration. Therefore, if the sample is not representative for the child population at risk, the results are meaningless. The authors need to explain how the sufficient sample size was determined.

We appreciate the reviewer’s concern on these two issues: sample size and representativeness, which we originally mentioned as a possible limitation of our study. As we explained, by pruning the original sample size of 41 children, we insured a more representative sample: more than half were excluded because their benzene metabolite levels were below the limit of detection and this was not realistic for our shoe-maker study population and would not work for the model used. Besides, we did not base the model solely on 15 children, the model already considered previously established parameters of 6-12 year children and these parameters were adjusted based on the characteristics of the 15 local children with exposure measurements. The following text was edited in the limitations section of the Discussion to make it clearer.

The children included in the model might not be representative of all the benzene-exposed children, but excluding more than half of the original 41 children increased the probability of representativeness, and we have no reason to believe that the remaining 15 are any different from all shoe-maker family children in terms of their biokinetic parameters. Furthermore, the characterization and validation of the PBPK model evaluates data such as formulas and pharmacokinetic and physiological parameters;  the original study design was a quota model and according to the guidelines of the Secretary of Health, for research in humans and in drug bioavailability studies, the number of individuals must be ≥ 12 (NOM-117-SSA1-2013)[44], [45].

2. Another thing that has to be described separately is the sampling strategy. I understand that this study relies on the earlier collected data, but these issues need to be addressed. Otherwise, this manuscript is nothing but a modeling exercise.

The sampling strategy of the original study from which we base our exposure assessment was described in the cited paper by Pérez-Herrera et al., 2019. Following the reviewer’s comment, we have complemented the description with the following explanation in section 2.3 Exposure assessment:

In order to compare the study population’s benzene exposure with benzene’s reference concentration for a risk assessment, it was necessary to use the concentration of benzene in the air. Since air benzene measurements were not available for our population, we estimated them from the urine tt-MA concentrations found in a previous study of children in Ticul (2019) [18];public invitations were made to parents of school-age children attending one of the three schools closest to the sites with the greatest environmental risks to public health including footwear manufacturing, agricultural and municipal garbage areas. The aforementioned investigation included a sample of 6- to 12-year-old children of shoemakers, potters or farmers family.

Reviewer 2 Report

I felt that the article was well written. Some editorial comments are given below. 

- Page 2, line 50: Remove “ “ (space) after “AML”.

- Page 3, Figure 1: Please add a scale for the right figure.

- Page 4, Lines 174-176: Move the definitions of the parameters below the equation 1 (as done for equation 2).

- Pages 4 & 5, equations 1 to 4: Normally an equation number appear after the equation. Please cheche the Journal format and correct them as necessary.

- Page 9, Lines 401-402: I wonder if it is appropriate to show the supplemental table here (just after Conclusion). Please check the Journal format and move the position as necessary.

Author Response

Reviewer 2

I felt that the article was well written. Some editorial comments are given below. 

Thank you, we address each comment below.

  1. Page 2, line 50: Remove “ “ (space) after “AML”.

The space was removed, thank you.

  1. Page 3, Figure 1: Please add a scale for the right figure.

We have added a scale for the map of Ticul’s workshops in Figure 1.

  1. Page 4, Lines 174-176: Move the definitions of the parameters below the equation 1 (as done for equation 2).

The definitions of the parameters were moved below the equation.

Pages 4 & 5, equations 1 to 4: Normally an equation number appear after the equation. Please cheche the Journal format and correct them as necessary.

The equation numbers were placed after the equation.

  1. Page 9, Lines 401-402: I wonder if it is appropriate to show the supplemental table here (just after Conclusion). Please check the Journal format and move the position as necessary.

We have now moved the supplementary table to another file so it can be available online.